# Dualing GANs

**Yujia Li**[1][*]  **Alexander Schwing**[3]  **Kuan-Chieh Wang**[1,2]  **Richard Zemel**[1,2]
[1]Department of Computer Science, University of Toronto  [2]Vector Institute
[3]Department of Electrical and Computer Engineering, University of Illinois at Urbana-Champaign
{yujiali, wangkua1, zemel}@cs.toronto.edu   aschwing@illinois.edu

## Abstract

Generative adversarial nets (GANs) are a promising technique for modeling a distribution from samples. It is however well known that GAN training suffers from instability due to the nature of its saddle point formulation. In this paper, we explore ways to tackle the instability problem by dualizing the discriminator. We start from linear discriminators in which case conjugate duality provides a mechanism to reformulate the saddle point objective into a maximization problem, such that both the generator and the discriminator of this 'dualing GAN' act in concert. We then demonstrate how to extend this intuition to non-linear formulations. For GANs with linear discriminators our approach is able to remove the instability in training, while for GANs with nonlinear discriminators our approach provides an alternative to the commonly used GAN training algorithm.

## 1  Introduction

Generative adversarial nets (GANs) [5] are, among others like variational auto-encoders [10] and auto-regressive models [19], a promising technique for modeling a distribution from samples. A lot of empirical evidence shows that GANs are able to learn to generate images with good visual quality at unprecedented resolution [22, 17], and recently there has been a lot of research interest in GANs, to better understand their properties and the training process.

Training GANs can be viewed as a duel between a discriminator and a generator. Both players are instantiated as deep nets. The generator is required to produce realistic-looking samples that cannot be differentiated from real data by the discriminator. In turn, the discriminator does as good a job as possible to tell the samples apart from real data. Due to the complexity of the optimization problem, training GANs is notoriously hard, and usually suffers from problems such as mode collapse, vanishing gradient, and divergence. Moreover, the training procedures are very unstable and sensitive to hyper-parameters. Therefore, a number of techniques have been proposed to address these issues, some empirically justified [17, 18], and others more theoretically motivated [15, 1, 16, 23].

This tremendous amount of recent work, together with the wide variety of heuristics applied by practitioners, indicates that many questions regarding the properties of GANs are still unanswered. In this work we provide another perspective on the properties of GANs, aiming toward better training algorithms in some cases. Our study in this paper is motivated by the alternating gradient update between discriminator and generator, employed during training of GANs. This form of update is one source of instability, and it is known to diverge even for some simple problems [18]. Ideally, when the discriminator is optimized to optimality, the GAN objective is a deterministic function of the generator. In this case, the optimization problem would be much easier to solve. This motivates our idea to dualize parts of the GAN objective, offering a mechanism to better optimize the discriminator.

Interestingly, our dual formulation provides a direct relationship between the GAN objective and the maximum mean-discrepancy framework discussed in [6]. When restricted to linear discriminators, where we can find the optimal discriminator by solving the dual, this formulation permits the derivation of an optimization algorithm that monotonically increases the objective. Moreover, for

---

[*]Now at DeepMind.

non-linear discriminators we can apply trust-region type optimization techniques to obtain more accurate discriminators. Our work brings to the table some additional optimization techniques beyond stochastic gradient descent; we hope this encourages other researchers to pursue this direction.

## 2 Background

In generative training we are interested in modeling of and sampling from an unknown distribution $P$, given a set $\mathcal{D} = \{\mathbf{x}_1, \ldots, \mathbf{x}_N\} \sim P$ of datapoints, for example images. GANs use a *generator* network $G_\theta(\mathbf{z})$ parameterized by $\theta$, that maps samples $\mathbf{z}$ drawn from a simple distribution, *e.g.*, Gaussian or uniform, to samples in the data space $\hat{\mathbf{x}} = G_\theta(\mathbf{z})$. A separate *discriminator* $D_\mathbf{w}(\mathbf{x})$ parameterized by $\mathbf{w}$ maps a point $\mathbf{x}$ in the data space to the probability of it being a real sample.

The discriminator is trained to minimize a classification loss, typically the cross-entropy, and the generator is trained to maximize the same loss. On sets of real data samples $\{\mathbf{x}_1, ..., \mathbf{x}_n\}$ and noise samples $\{\mathbf{z}_1, ..., \mathbf{z}_n\}$, using the (averaged) cross-entropy loss results in the following joint optimization problem:

$$\max_\theta \min_\mathbf{w} f(\theta, \mathbf{w}) \quad \text{where} \quad f(\theta, \mathbf{w}) = -\frac{1}{2n} \sum_i \log D_\mathbf{w}(\mathbf{x}_i) - \frac{1}{2n} \sum_i \log(1 - D_\mathbf{w}(G_\theta(\mathbf{z}_i))). \quad (1)$$

We adhere to the formulation of a fixed batch of samples for clarity of the presentation, but also point out how this process is adapted to the stochastic optimization setting later in the paper as well as in the supplementary material.

To solve this saddle point optimization problem, ideally, we want to solve for the optimal discriminator parameters $\mathbf{w}^*(\theta) = \operatorname{argmin}_\mathbf{w} f(\theta, \mathbf{w})$, in which case the GAN program given in Eq. (1) can be reformulated as a maximization for $\theta$ using $\max_\theta f(\theta, \mathbf{w}^*(\theta))$. However, typical GAN training only alternates two gradient updates $\mathbf{w} \leftarrow \mathbf{w} - \eta_\mathbf{w} \nabla_\mathbf{w} f(\theta, \mathbf{w})$ and $\theta \leftarrow \theta + \eta_\theta \nabla_\theta f(\theta, \mathbf{w})$, and usually just one step for each of $\theta$ and $\mathbf{w}$ in each round. In this case, the objective maximized by the generator is $f(\theta, \mathbf{w})$ instead. This objective is always an upper bound on the correct objective $f(\theta, \mathbf{w}^*(\theta))$, since $\mathbf{w}^*(\theta)$ is the optimal $\mathbf{w}$ for $\theta$. Maximizing an upper bound has no guarantee on maximizing the correct objective, which leads to instability. Therefore, many practically useful techniques have been proposed to circumvent the difficulties of the original program definition presented in Eq. (1).

Another widely employed technique is a separate loss $-\sum_i \log(D_\mathbf{w}(G_\theta(\mathbf{z}_i)))$ to update $\theta$ in order to avoid vanishing gradients during early stages of training when the discriminator can get too strong. This technique can be combined with our approach, but in what follows, we keep the elegant formulation of the GAN program specified in Eq. (1).

## 3 Dualing GANs

The main idea of 'Dualing GANs' is to represent the discriminator program $\min_\mathbf{w} f(\theta, \mathbf{w})$ in Eq. (1) using its dual, $\max_\lambda g(\theta, \lambda)$. Hereby, $g$ is the dual objective of $f$ w.r.t. $\mathbf{w}$, and $\lambda$ are the dual variables. Instead of gradient descent on $f$ to update $\mathbf{w}$, we solve the dual instead. This results in a maximization problem $\max_\theta \max_\lambda g(\theta, \lambda)$.

Using the dual is beneficial for two reasons. First, note that for any $\lambda$, $g(\theta, \lambda)$ is a lower bound on the objective with optimal discriminator parameters $f(\theta, \mathbf{w}^*(\theta))$. Staying in the dual domain, it is then guaranteed that optimization of $g$ w.r.t. $\theta$ makes progress in terms of the original program. Second, the dual problem usually involves a much smaller number of variables, and can therefore be solved much more easily than the primal formulation. This provides opportunities to obtain more accurate estimates for the discriminator parameters $\mathbf{w}$, which is in turn beneficial for stabilizing the learning of the generator parameters $\theta$. In the following, we start by studying linear discriminators, before extending our technique to training with non-linear discriminators. Also, we use cross-entropy as the classification loss, but emphasize that other convex loss functions, *e.g.*, the hinge-loss, can be applied equivalently.

### 3.1 Linear Discriminator

We start from linear discriminators that use a linear scoring function $F(\mathbf{w}, \mathbf{x}) = \mathbf{w}^\top \mathbf{x}$, *i.e.*, the discriminator $D_\mathbf{w}(\mathbf{x}) = p_\mathbf{w}(y = 1|\mathbf{x}) = \sigma(F(\mathbf{w}, \mathbf{x})) = 1/[1 + \exp(-\mathbf{w}^\top \mathbf{x})]$. Here, $y = 1$ indicates real data, while $y = -1$ for a generated sample. The distribution $p_\mathbf{w}(y = -1|\mathbf{x}) = 1 - p_\mathbf{w}(y = 1|\mathbf{x})$ characterizes the probability of $\mathbf{x}$ being a generated sample.

We only require the scoring function $F$ to be linear in $\mathbf{w}$ and any (nonlinear) differentiable features $\phi(\mathbf{x})$ can be used in place of $\mathbf{x}$ in this formulation. Substituting the linear scoring function into the objective given in Eq. (1), results in the following program for $\mathbf{w}$:

$$\min_{\mathbf{w}} \quad \frac{C}{2}\|\mathbf{w}\|_2^2 + \frac{1}{2n}\sum_i \log(1 + \exp(-\mathbf{w}^\top \mathbf{x}_i)) + \frac{1}{2n}\sum_i \log(1 + \exp(\mathbf{w}^\top G_\theta(\mathbf{z}_i))). \quad (2)$$

Here we also added an L2-norm regularizer on $\mathbf{w}$. We note that the program presented in Eq. (2) is convex in the discriminator parameters $\mathbf{w}$. Hence, we can equivalently solve it in the dual domain as discussed in the following claim, with proof provided in the supplementary material.

**Claim 1.** *The dual program to the task given in Eq.* (2) *reads as follows:*

$$\max_{\lambda} \quad g(\theta, \lambda) = -\frac{1}{2C}\left\|\sum_i \lambda_{\mathbf{x}_i}\mathbf{x}_i - \sum_i \lambda_{\mathbf{z}_i}G_\theta(\mathbf{z}_i)\right\|^2 + \frac{1}{2n}\sum_i H(2n\lambda_{\mathbf{x}_i}) + \frac{1}{2n}\sum_i H(2n\lambda_{\mathbf{z}_i}),$$

$$\text{s.t.} \quad \forall i, \quad 0 \le \lambda_{\mathbf{x}_i} \le \frac{1}{2n}, \quad 0 \le \lambda_{\mathbf{z}_i} \le \frac{1}{2n}, \quad\quad\quad\quad\quad (3)$$

*with binary entropy $H(u) = -u\log u - (1-u)\log(1-u)$. The optimal solution to the original problem $\mathbf{w}^*$ can be obtained from the optimal $\lambda_{\mathbf{x}_i}^*$ and $\lambda_{\mathbf{z}_i}^*$ via*

$$\mathbf{w}^* = \frac{1}{C}\left(\sum_i \lambda_{\mathbf{x}_i}^*\mathbf{x}_i - \sum_i \lambda_{\mathbf{z}_i}^*G_\theta(\mathbf{z}_i)\right).$$

**Remarks:** Intuitively, considering the last two terms of the program given in Claim 1 as well as its constraints, we aim at assigning weights $\lambda_{\mathbf{x}}, \lambda_{\mathbf{z}}$ close to half of $\frac{1}{2n}$ to as many data points and to as many artificial samples as possible. More carefully investigating the first part, which can at most reach zero, reveals that we aim to match the empirical data observation $\sum_i \lambda_{\mathbf{x}_i}\mathbf{x}_i$ and the generated artificial sample observation $\sum_i \lambda_{\mathbf{z}_i}G_\theta(\mathbf{z}_i)$. Note that this resembles the moment matching property obtained in other maximum likelihood models. Importantly, this objective also resembles the (kernel) maximum mean discrepancy (MMD) framework, where the empirical squared MMD is estimated via $\|\frac{1}{n}\sum_{\mathbf{x}_i}\mathbf{x}_i - \frac{1}{n}\sum_{\mathbf{z}_i}G_\theta(\mathbf{z}_i)\|_2^2$. Generative models that learn to minimize the MMD objective, like the generative moment matching networks [13, 3], can therefore be included in our framework, using fixed $\lambda$'s and proper scaling of the first term.

Combining the result obtained in Claim 1 with the training objective for the generator yields the task $\max_{\theta,\lambda} g(\theta, \lambda)$ for training of GANs with linear discriminators. Hence, instead of searching for a saddle point, we strive to find a maximizer, a task which is presumably easier. The price to pay is the restriction to linear discriminators and the fact that every randomly drawn artificial sample $\mathbf{z}_i$ has its own dual variable $\lambda_{\mathbf{z}_i}$.

In the non-stochastic optimization setting, where we optimize for fixed sets of data samples $\{\mathbf{x}_i\}$ and randomizations $\{\mathbf{z}_i\}$, it is easy to design a learning algorithm for GANs with linear discriminators that monotonically improves the objective $g(\theta, \lambda)$ based on line search. Although this approach is not practical for very large data sets, such a property is convenient for smaller scale data sets. In addition, linear models are favorable in scenarios in which we know informative features that we want the discriminator to pay attention to.

When optimizing with mini-batches we introduce new data samples $\{\mathbf{x}_i\}$ and randomizations $\{\mathbf{z}_i\}$ in every iteration. In the supplementary material we show that this corresponds to maximizing a lower bound on the full expectation objective. Since the dual variables vary from one mini-batch to the next, we need to solve for the newly introduced dual variables to a reasonable accuracy. For small minibatch sizes commonly used in deep learning literature, like 100, calling a constrained optimization solver to solve the dual problem is quite cheap. We used Ipopt [20], which typically solves this dual problem to a good accuracy in negligible time; other solvers can also be used and may lead to improved performance.

Utilizing a log-linear discriminator reduces the model's expressiveness and complexity. We therefore now propose methods to alleviate this restriction.

## 3.2 Non-linear Discriminator

General non-linear discriminators use non-convex scoring functions $F(\mathbf{w}, \mathbf{x})$, parameterized by a deep net. The non-convexity of $F$ makes it hard to directly convert the problem into its dual form.

Therefore, our approach for training GANs with non-convex discriminators is based on repeatedly linearizing and dualizing the discriminator locally. At first sight this seems restrictive, however, we will show that a specific setup of this technique recovers the gradient direction employed in the regular GAN training mechanism while providing additional flexibility.

We consider locally approximating the primal objective $f$ around a point $\mathbf{w}_k$ using a model function $m_{k,\theta}(\mathbf{s}) \approx f(\theta, \mathbf{w}_k + \mathbf{s})$. We phrase the update w.r.t. the discriminator parameters $\mathbf{w}$ as a search for a step $\mathbf{s}$, i.e., $\mathbf{w}_{k+1} = \mathbf{w}_k + \mathbf{s}$ where $k$ indicates the current iteration. In order to guarantee the quality of the approximation, we introduce a trust-region constraint $\frac{1}{2}\|\mathbf{s}\|_2^2 \leq \Delta_k \in \mathbb{R}^+$ where $\Delta_k$ specifies the trust-region size. More concretely, we search for a step $\mathbf{s}$ by solving

$$\min_{\mathbf{s}} m_{k,\theta}(\mathbf{s}) \quad \text{s.t.} \quad \frac{1}{2}\|\mathbf{s}\|_2^2 \leq \Delta_k, \tag{4}$$

given generator parameters $\theta$. Rather than optimizing the GAN objective $f(\theta, \mathbf{w})$ with stochastic gradient descent, we can instead employ this model function and use the algorithm outlined in Alg. 1. It proceeds by first performing a gradient ascent w.r.t. the generator parameters $\theta$. Afterwards, we find a step $\mathbf{s}$ by solving the program given in Eq. (4). We then apply this step, and repeat.

Different model functions $m_{k,\theta}(\mathbf{s})$ result in variants of the algorithm. If we choose $m_{k,\theta}(\mathbf{s}) = f(\theta, \mathbf{w}_k + \mathbf{s})$, model $m$ and function $f$ are identical but the program given in Eq. (4) is hard to solve. Therefore, in the following, we propose two model functions that we have found to be useful. The first one is based on linearization of the cost function $f(\theta, \mathbf{w})$ and recovers the step $\mathbf{s}$ employed by gradient-based discriminator updates in standard GAN training. The second one is based on linearization of the score function $F(\mathbf{w}, \mathbf{x})$ while keeping the loss function intact; this second approximation is hence accurate in a larger region. Many more models $m_{k,\theta}(\mathbf{s})$ exist and we leave further exploration of this space to future work.

**(A). Cost function linearization:** A local approximation to the cost function $f(\theta, \mathbf{w})$ can be constructed by using the first order Taylor approximation

$$m_{k,\theta}(\mathbf{s}) = f(\mathbf{w}_k, \theta) + \nabla_{\mathbf{w}} f(\mathbf{w}_k, \theta)^\top \mathbf{s}.$$

Such a model function is appealing because step 2 of Fig. 1, i.e., minimization of the model function subject to trust-region constraints as specified in Eq. (4), has the analytically computable solution

$$\mathbf{s} = -\frac{\sqrt{2\Delta_k}}{\|\nabla_{\mathbf{w}} f(\mathbf{w}_k, \theta)\|_2} \nabla_{\mathbf{w}} f(\mathbf{w}_k, \theta).$$

Consequently step 3 of Fig. 1 is a step of length $2\Delta_k$ into the negative gradient direction of the cost function $f(\theta, \mathbf{w})$. We can use the trust region parameter $\Delta_k$ to tune the step size just like it is common to specify the step size for standard GAN training. As mentioned before, using the first order Taylor approximation as our model $m_{k,\theta}(\mathbf{s})$ recovers the same direction that is employed during standard GAN training. The value of the $\Delta_k$ parameters can be fixed or adapted; see the supplementary material for more details.

Using the first order Taylor approximation as a model is not the only choice. While some choices like quadratic approximation are fairly obvious, we present another intriguing option in the following.

**(B). Score function linearization:** Instead of linearizing the entire cost function as demonstrated in the previous part, we can choose to only linearize the score function $F$, locally around $\mathbf{w}_k$, via

$$F(\mathbf{w}_k + \mathbf{s}, \mathbf{x}) \approx \hat{F}(\mathbf{s}, \mathbf{x}) = F(\mathbf{w}_k, \mathbf{x}) + \mathbf{s}^\top \nabla_{\mathbf{w}} F(\mathbf{w}_k, \mathbf{x}), \quad \forall \mathbf{x}.$$

Note that the overall objective $f$ is itself a nonlinear function of $F$. Substituting the approximation for $F$ into the overall objective, results in the following model function:

$$m_{k,\theta}(\mathbf{s}) = \frac{C}{2}\|\mathbf{w}_k + \mathbf{s}\|_2^2 + \frac{1}{2n}\sum_i \log\left(1 + \exp\left(-F(\mathbf{w}_k, \mathbf{x}_i) - \mathbf{s}^\top \nabla_{\mathbf{w}} F(\mathbf{w}_k, \mathbf{x}_i)\right)\right)$$

$$+ \frac{1}{2n}\sum_i \log\left(1 + \exp\left(F(\mathbf{w}_k, G_\theta(\mathbf{z}_i)) + \mathbf{s}^\top \nabla_{\mathbf{w}} F(\mathbf{w}_k, G_\theta(\mathbf{z}_i))\right)\right). \tag{5}$$

This approximation keeps the nonlinearities of the surrogate loss function intact, therefore we expect it to be more accurate than linearization of the whole cost function $f(\theta, \mathbf{w})$. When $F$ is already linear in $\mathbf{w}$, linearization of the score function introduces no approximation error, and the formulation can be naturally reduced to the discussion presented in Sec. 3.1; non-negligible errors are introduced when linearizing the whole cost function $f$ in this case.

---

**Algorithm 1** GAN optimization with model function.

---

Initialize $\theta$, $\mathbf{w}_0$, $k = 0$ and iterate

1. One or few gradient ascent steps on $f(\theta, \mathbf{w}_k)$ w.r.t. generator parameters $\theta$
2. Find step $\mathbf{s}$ using $\min_{\mathbf{s}} m_{k,\theta}(\mathbf{s})$ s.t. $\frac{1}{2}\|\mathbf{s}\|_2^2 \leq \Delta_k$
3. Update $\mathbf{w}_{k+1} \leftarrow \mathbf{w}_k + \mathbf{s}$
4. $k \leftarrow k + 1$

---

For general non-linear discriminators, however, no analytic solution can be computed for the program given in Eq. (4) when using this model. Nonetheless, the model function fulfills $m_{k,\theta}(0) = f(\mathbf{w}_k, \theta)$ and it is convex in $\mathbf{s}$. Exploiting this convexity, we can derive the dual for this trust-region optimization problem as presented in the following claim. The proof is included in the supplementary material.

**Claim 2.** *The dual program to* $\min_{\mathbf{s}} m_{k,\theta}(\mathbf{s})$ *s.t.* $\frac{1}{2}\|\mathbf{s}\|_2^2 \leq \Delta_k$ *with model function as in Eq. (5) is:*

$$
\max_{\lambda} \quad \frac{C}{2}\|\mathbf{w}_k\|_2^2 - \frac{1}{2(C + \lambda_T)}\left\|-C\mathbf{w}_k + \sum_i \lambda_{\mathbf{x}_i} \nabla_{\mathbf{w}} F(\mathbf{w}_k, \mathbf{x}_i) - \sum_i \lambda_{\mathbf{z}_i} \nabla_{\mathbf{w}} F(\mathbf{w}_k, G_\theta(\mathbf{z}_i))\right\|_2^2
$$

$$
+ \frac{1}{2n}\sum_i H(2n\lambda_{\mathbf{x}_i}) + \frac{1}{2n}\sum_i H(2n\lambda_{\mathbf{z}_i}) - \sum_i \lambda_{\mathbf{x}_i} F_{\mathbf{x}_i} + \sum_i \lambda_{\mathbf{z}_i} F_{\mathbf{z}_i} - \lambda_T \Delta_k
$$

$$
\text{s.t.} \quad \lambda_T \geq 0 \qquad \forall i, \quad 0 \leq \lambda_{\mathbf{x}_i} \leq \frac{1}{2n}, \quad 0 \leq \lambda_{\mathbf{z}_i} \leq \frac{1}{2n}.
$$

*The optimal* $\mathbf{s}^*$ *to the original problem can be expressed through optimal* $\lambda_T^*, \lambda_{\mathbf{x}_i}^*, \lambda_{\mathbf{z}_i}^*$ *as*

$$
\mathbf{s}^* = \frac{1}{C + \lambda_T^*}\left(\sum_i \lambda_{\mathbf{x}_i}^* \nabla_{\mathbf{w}} F(\mathbf{w}_k, \mathbf{x}_i) - \sum_i \lambda_{\mathbf{z}_i}^* \nabla_{\mathbf{w}} F(\mathbf{w}_k, \mathbf{z}_i)\right) - \frac{C}{C + \lambda_T^*}\mathbf{w}_k
$$

Combining the dual formulation with the maximization of the generator parameters $\theta$ results in a maximization as opposed to a search for a saddle point. However, unlike the linear case, it is not possible to design an algorithm that is guaranteed to monotonically increase the cost function $f(\theta, \mathbf{w})$. The culprit is step 3 of Alg. 1, which adapts the model $m_{k,\theta}(\mathbf{s})$ in every iteration.

Intuitively, the program illustrated in Claim 2 aims at choosing dual variables $\lambda_{\mathbf{x}_i}, \lambda_{\mathbf{z}_i}$ such that the weighted means of derivatives as well as scores match. Note that this program searches for a direction $\mathbf{s}$ as opposed to searching for the weights $\mathbf{w}$, hence the term $-C\mathbf{w}_k$ inside the squared norm.

In practice, we use Ipopt [20] to solve the dual problem. The form of this dual is more ill-conditioned than the linear case. The solution found by Ipopt sometimes contains errors, however, we found the errors to be generally tolerable and not to affect the performance of our models.

## 4 Experiments

In this section, we empirically study the proposed dual GAN algorithms. In particular, we show the stable and monotonic training for linear discriminators and study its properties. For nonlinear GANs we show good quality samples and compare it with standard GAN training methods. Overall the results show that our proposed approaches work across a range of problems and provide good alternatives to the standard GAN training method.

### 4.1 Dual GAN with linear discriminator

We explore the dual GAN with linear discriminator on a synthetic 2D dataset generated by sampling points from a mixture of 5 2D Gaussians, as well as the MNIST [12] dataset. Through these experiments we show that (1) with the proposed dual GAN algorithm, training is very stable; (2) the dual variables $\lambda$ can be used as an extra informative signal for monitoring the training process; (3) features matter, and we can train good generative models even with linear discriminators when we have good features. In all experiments, we compare our proposed dual GAN with the standard GAN when training the same generator and discriminator models. Additional experimental details and results are included in the supplementary material.

The discussion of linear discriminators presented in Sec. 3.1 works with any feature representation $\phi(\mathbf{x})$ in place of $\mathbf{x}$ as long as $\phi$ is differentiable to allow gradients flow through it. For the simple

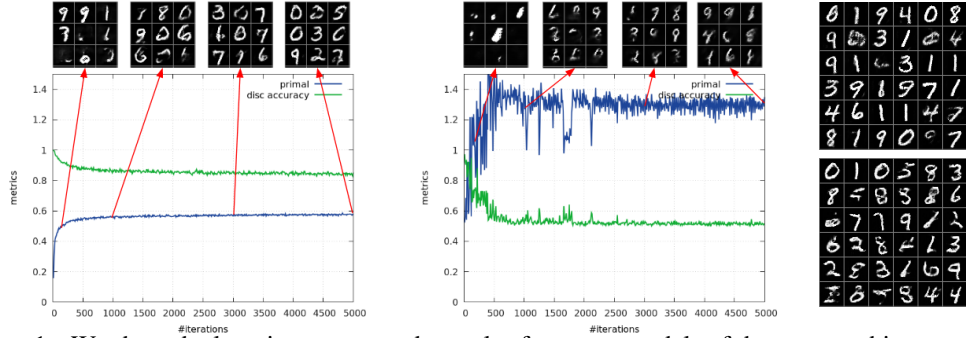

Figure 1: We show the learning curves and samples from two models of the same architecture, one optimized in dual space (left), and one in the primal space (*i.e.*, typical GAN) up to 5000 iterations. Samples are shown at different points during training, as well as at the very end (right top - dual, right bottom - primal). Despite having similar sample qualities in the end, they demonstrate drastically different training behavior. In the typical GAN setup, loss oscillates and has no clear trend, whereas in the dual setup, loss monotonically increases and shows much smaller oscillation. Sample quality is nicely correlated with the dual objective during training.

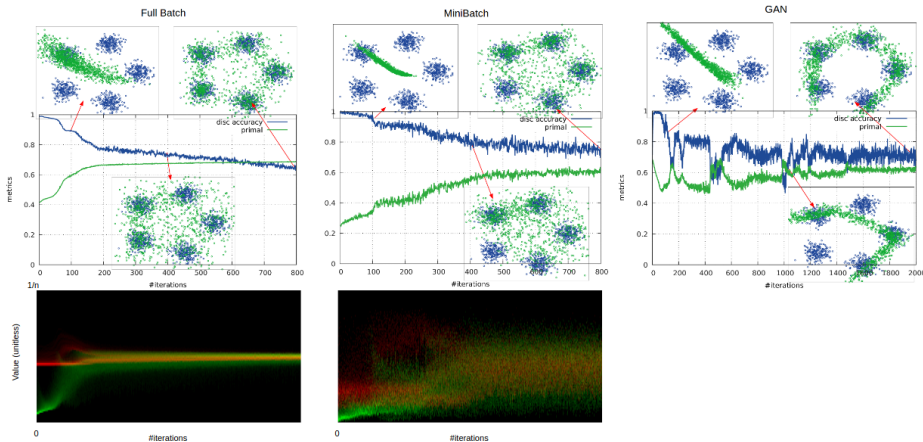

Figure 2: Training GANs with linear discriminators on the simple 5-Gaussians dataset. Here we are showing typical runs with the compared methods (not cherry-picked). Top: training curves and samples from a single experiment: left - dual with full batch, middle - dual with minibatch, right - standard GAN with minibatch. The real data from this dataset are drawn in blue, generated samples in green. Below: distribution of $\lambda$'s during training for the two dual GAN experiments, as a histogram at each x-value (iteration) where intensity depicts frequency for values ranging from 0 to 1 (red are data, and green are samples).

5-Gaussian dataset, we use RBF features based on 100 sample training points. For the MNIST dataset, we use a convolutional neural net, and concatenate the hidden activations on all layers as the features.

The dual GAN formulation has a single hyper-parameter $C$, but we found the algorithm not to be sensitive to it, and set it to 0.0001 in all experiments. We used Adam [9] with fixed learning rate and momentum to optimize the generator.

**Stable Training:** The main results illustrating stable training are provided in Fig. 1 and 2, where we show the learning curves as well as model samples at different points during training. Both the dual GAN and the standard GAN use minibatches of the same size, and for the synthetic dataset we did an extra experiment doing full-batch training. From these curves we can see the stable monotonic increase of the dual objective, contrasted with standard GAN's spiky training curves. On the synthetic data, we see that increasing the minibatch size leads to significantly improved stability. In the supplementary material we include an extra experiment to quantify the stability of the proposed method on the synthetic dataset.

| Dataset | mini-batch size | generator learnrate | generator momentum | $C$ | discriminator learnrate* | generator architecture | max iterations |
|---|---|---|---|---|---|---|---|
| 5-Gaussians | randint[20,200] | enr([0,10]) | rand[.1,.9] | enr([0,6]) | enr([0,10]) | fc-small<br>fc-large | randint[400,2000] |
| MNIST | randint[20,200] | enr([0,10]) | rand[.1,.9] | enr([0,6]) | enr([0,10]) | fc-small<br>fc-large<br>dcgan<br>dcgan-no-bn | 20000 |

Table 1: Ranges of hyperparameters for sensitivity experiment. randint[a,b] means samples were drawn from uniformly distributed integers in the closed interval of [a,b], similarly rand[a,b] for real numbers. enr([a,b]) is shorthand for exp(-randint[a,b]), which was used for hyperparameters commonly explored in log-scale. For generator architectures, for the 5-Gaussians dataset we tried 2 3-layer fully-connected networks, with 20 and 40 hidden units. For MNIST, we tried 2 3-layer fully-connected networks, with 256 and 1024 hidden units, and a DCGAN-like architecture with and without batch normalization.

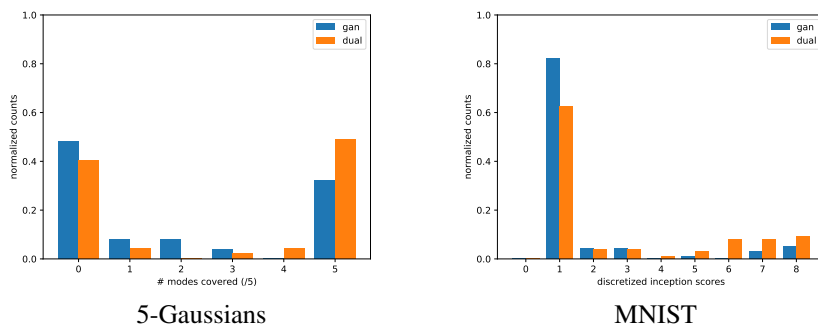

| 5-Gaussians | MNIST |
|---|---|

Figure 3: Results for hyperparameter sensitivity experiment. For 5-Gaussians dataset, the x-axis represents the number of modes covered. For MNIST, the x-axis represents discretized Inception score. Overall, the proposed dual GAN results concentrate significantly more mass on the right side, demonstrating its better robustness to hyperparameters than standard GANs.

**Sensitivity to Hyperparameters:** Sensitivity to hyperparameters is another important aspect of training stability. Successful GAN training typically requires carefully tuned hyperparameters, making it difficult for non-experts to adopt these generative models. In an attempt to quantify this sensitivity, we investigated the robustness of the proposed method to the hyperparameter choice. For both the 5-Gaussians and MNIST datasets, we randomly sampled 100 hyperparameter settings from ranges specified in Table 1, and compared learning using both the proposed dual GAN and the standard GAN. On the 5-Gaussians dataset, we evaluated the performance of the models by how well the model samples covered the 5 modes. We defined successfully covering a mode as having $> 100$ out of 1000 samples falling within a distance of 3 standard deviations to the center of the Gaussian. Our dual linear GAN succeeded in 49% of the experiments (note that there are a significant number of bad hyperparameter combinations in the search range), and standard GAN succeeded in only 32%, demonstrating our method was significantly easier to train and tune. On MNIST, the mean Inception scores were 2.83, 1.99 for the proposed method and GAN training respectively. A more detailed breakdown of mode coverage and Inception score can be found in Figure 3.

**Distribution of $\lambda$ During Training:** The dual formulation allows us to monitor the training process through a unique perspective by monitoring the dual variables $\lambda$. Fig. 2 shows the evolution of the distribution of $\lambda$ during training for the synthetic 2D dataset. At the begining of training the $\lambda$'s are on the low side as the generator is not good and $\lambda$'s are encouraged to be small to minimize the moment matching cost. As the generator improves, more attention is devoted to the entropy term in the dual objective, and the $\lambda$'s start to converge to the value of $1/(4n)$.

**Comparison of Different Features:** The qualitative differences of the learned models with different features can be observed in Fig. 4. In general, the more information the features carry about the data, the better the learned generative models. On MNIST, even with random features and linear discriminators we can learn reasonably good generative models. On the other hand, these results also

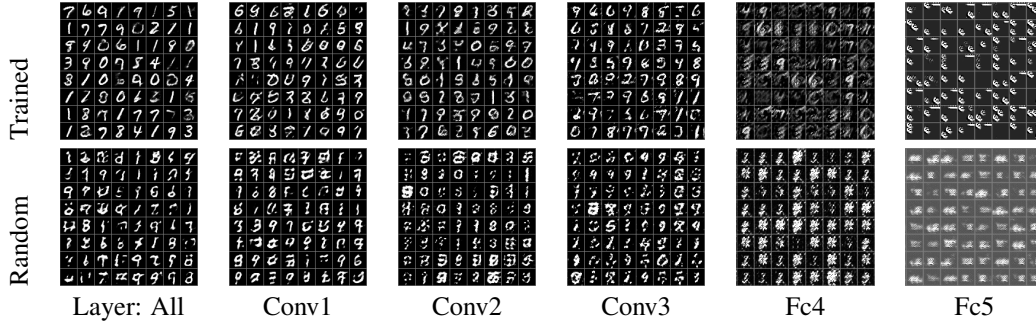

| | | | | | |
| Trained | | | | | |
| Random | | | | | |
| Layer: All | Conv1 | Conv2 | Conv3 | Fc4 | Fc5 |

Figure 4: Samples from dual linear GAN using pretrained and random features on MNIST. Each column shows a set of different features, utilizing all layers in a convnet and then successive single layers in the network.

| Score Type | GAN | Score Lin | Cost Lin | Real Data |
|---|---|---|---|---|
| Inception (end) | 5.61±0.09 | 5.40±0.12 | 5.43±0.10 | 10.72 ± 0.38 |
| Internal classifier (end) | 3.85±0.08 | 3.52±0.09 | 4.42±0.09 | 8.03 ± 0.07 |
| Inception (avg) | 5.59±0.38 | 5.44±0.08 | 5.16±0.37 | - |
| Internal classifier (avg) | 3.64±0.47 | 3.70±0.27 | 4.04±0.37 | - |

Table 2: Inception Score [18] for different GAN training methods. Since the score depends on the classifier, we used code from [18] as well as our own small convnet CIFAR-10 classifier for evaluation (achieves 83% accuracy). All scores are computed using 10,000 samples. The top pair are scores on the final models. GANs are known to be unstable, and results are sometimes cherry-picked. So, the bottom pair are scores averaged across models sampled from different iterations of training after it stopped improving.

indicate that if the features are bad then it is hard to learn good models. This leads us to the nonlinear discriminators presented below, where the discriminator features are learned together with the last layer, which may be necessary for more complicated problem domains where features are potentially difficult to engineer.

## 4.2 Dual GAN with non-linear discriminator

Next we assess the applicability of our proposed technique for non-linear discriminators, and focus on training models on MNIST and CIFAR-10 [11].

As discussed in Sec. 3.2, when the discriminator is non-linear, we can only approximate the discriminator locally. Therefore we do not have monotonic convergence guarantees. However, through better approximation and optimization of the discriminator we may expect the proposed dual GAN to work better than standard gradient based GAN training in some cases. Since GAN training is sensitive to hyperparameters, to make the comparison fair, we tuned the parameters for both the standard GANs and our approaches extensively and compare the best results for each.

Fig. 5 and 6 show the samples generated by models learned using different approaches. Visually samples of our proposed approaches are on par with the standard GANs. As an extra quantitative metric for performance, we computed the Inception Score [18] for each of them on CIFAR-10 in Table 2. The Inception Score is a surrogate metric which highly depends on the network architecture. Therefore we computed the score using our own classifier and the one proposed in [18]. As can be seen in Table 2, both score and cost linearization are competitive with standard GANs. From the training curves we can also see that score linearization does the best in terms of approximating the objective, and both score linearization and cost linearization oscillate less than standard GANs.

## 5 Related Work

A thorough review of the research devoted to generative modeling is beyond the scope of this paper. In this section we focus on GANs [5] and review the most related work that has not been discussed throughout the paper.

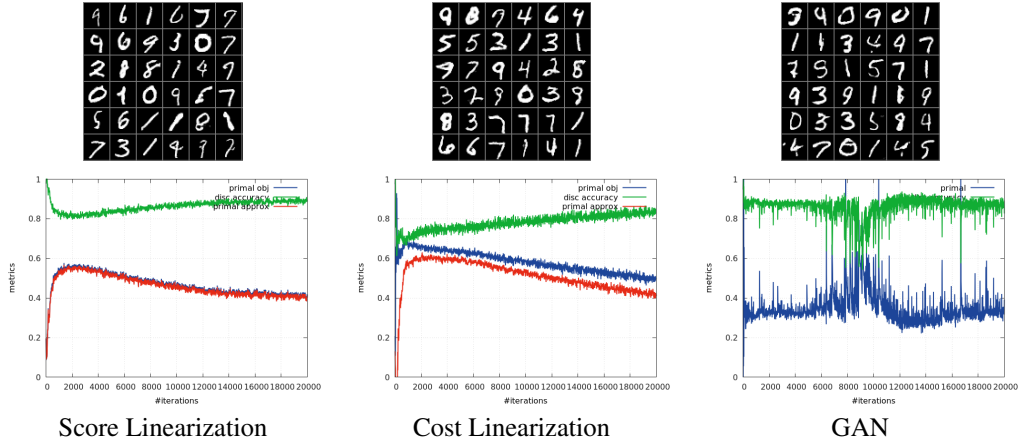

| Score Linearization | Cost Linearization | GAN |

Figure 5: Nonlinear discriminator experiments on MNIST, and their training curves, showing the primal objective, the approximation, and the discriminator accuracy. Here we are showing typical runs with the compared methods (not cherry-picked).

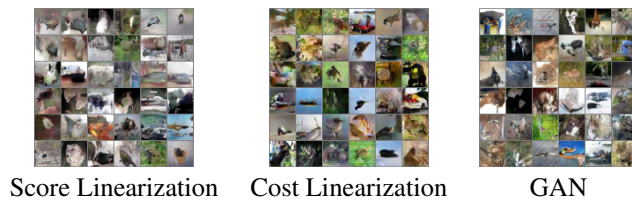

Score Linearization   Cost Linearization   GAN

Figure 6: Nonlinear discriminator experiments on CIFAR-10, learning curves and samples organized by class are provided in the supplementary material.

Our dual formulation reveals a close connection to moment-matching objectives widely seen in many other models. MMD [6] is one such related objective, and has been used in deep generative models in [13, 3]. [18] proposed a range of techniques to improve GAN training, including the usage of feature matching. Similar techniques are also common in style transfer [4]. In addition to these, moment-matching objectives are very common for exponential family models [21]. Common to all these works is the use of fixed moments. The Wasserstein objective proposed for GAN training in [1] can also be thought of as a form of moment matching, where the features are part of the discriminator and they are adaptive. The main difference between our dual GAN with linear discriminators and other forms of adaptive moment matching is that we adapt the weighting of features by optimizing non-parametric dual parameters, while other works mostly adopt a parametric model to adapt features.

Duality has also been studied to understand and improve GAN training. [16] pioneered work that uses duality to derive new GAN training objectives from other divergences. [1] also used duality to derive a practical objective for training GANs from other distance metrics. Compared to previous work, instead of coming up with new objectives, we instead used duality on the original GAN objective and aim to better optimize the discriminator.

Beyond what has already been discussed, there has been a range of other techniques developed to improve or extend GAN training, *e.g.*, [8, 7, 22, 2, 23, 14] just to name a few.

## 6 Conclusion

To conclude, we introduced 'Dualing GANs,' a framework which considers duality based formulations for the duel between the discriminator and the generator. Using the dual formulation provides opportunities to better train the discriminator. This helps remove the instability in training for linear discriminators, and we also adapted this framework to non-linear discriminators. The dual formulation also provides connections to other techniques. In particular, we discussed a close link to moment matching techniques, and showed that the cost function linearization for non-linear discriminators recovers the original gradient direction in standard GANs. We hope that our results spur further research in this direction to obtain a better understanding of the GAN objective and its intricacies.

**Acknowledgments:** This material is based upon work supported in part by the National Science Foundation under Grant No. 1718221, and grants from NSERC, Samsung and CIFAR.

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
