[Supplementary Material · supp.pdf]

# Supplementary Material for Dualing GANs

**Yujia Li**[1]* **Alexander Schwing**[2] **Kuan-Chieh Wang**[1] **Richard Zemel**[1]
[1]Department of Computer Science [2]Department of Electrical and Computer Engineering
University of Toronto University of Illinois at Urbana-Champaign
{yujiali, wangkua1, zemel}@cs.toronto.edu aschwing@illinois.edu

## A Minibatch objective

Standard GAN training are motivated from a maximin formulation on an expectation objective

$$\max_{\theta} \min_{\mathbf{w}} \mathbb{E}_{p_d(\mathbf{x})}[\log D_{\mathbf{w}}(\mathbf{x})] + \mathbb{E}_{p_z(\mathbf{z})}[\log(1 - D_{\mathbf{w}}(G_{\theta}(\mathbf{z})))] \tag{1}$$

where $p_d$ is the true data distribution, and $p_z$ is the prior distribution on $\mathbf{z}$.

In practice, however, a minibatch of data $\mathbf{X} = \{\mathbf{x}_1, ..., \mathbf{x}_n\}$ and noise $\mathbf{Z} = \{\mathbf{z}_1, ..., \mathbf{z}_n\}$ are sampled each time, and one gradient update is made to update $\mathbf{w}$ and $\theta$ each.

In our formulation, in particular the dual GAN with linear discriminators, we can solve the inner optimization problem over $\mathbf{w}$ on minibatch samples $\mathbf{X}$ and $\mathbf{Z}$ to optimality, $\theta$ is then updated with the optimal $\mathbf{w}$. This effectively makes the optimization problem take the following form

$$\max_{\theta} \min_{\mathbf{w}} \mathbb{E}_{\mathbf{X},\mathbf{Z}}[f(\mathbf{w}, \theta, \mathbf{X}, \mathbf{Z})], \quad \text{where} \quad f(\mathbf{w}, \theta, \mathbf{X}, \mathbf{Z}) = \frac{1}{n} \sum_i \hat{f}_x(\mathbf{w}, \theta, \mathbf{x}_i) + \frac{1}{n} \sum_i \hat{f}_z(\mathbf{w}, \theta, \mathbf{z}_i) \tag{2}$$

where $\hat{f}_x$ and $\hat{f}_z$ are the individual loss functions. Using this notation, the original GAN problem can be represented as

$$\max_{\theta} \min_{\mathbf{w}} \mathbb{E}_{\mathbf{x},\mathbf{z}}[f(\mathbf{w}, \theta, \{\mathbf{x}\}, \{\mathbf{z}\})] = \max_{\theta} \min_{\mathbf{w}} \mathbb{E}_{\mathbf{X},\mathbf{Z}}[f(\mathbf{w}, \theta, \mathbf{X}, \mathbf{Z})] \tag{3}$$

since, $\mathbf{x}_i$ and $\mathbf{z}_i$ are drawn i.i.d. from corresponding distributions.

Let $\mathbf{w}^* = \text{argmin}_{\mathbf{w}} \mathbb{E}_{\mathbf{X},\mathbf{Z}}[f(\mathbf{w}, \theta, \mathbf{X}, \mathbf{Z})]$, we have

$$\mathbb{E}_{\mathbf{X},\mathbf{Z}}[\min_{\mathbf{w}} f(\mathbf{w}, \theta, \mathbf{X}, \mathbf{Z})] \leq \mathbb{E}_{\mathbf{X},\mathbf{Z}}[f(\mathbf{w}^*, \theta, \mathbf{X}, \mathbf{Z})] = \min_{\mathbf{w}} \mathbb{E}_{\mathbf{X},\mathbf{Z}}[f(\mathbf{w}, \theta, \mathbf{X}, \mathbf{Z})], \tag{4}$$

which means our minibatch algorithm is actually optimizing a lower bound on the theoretical GAN objective, this introduces a bias that decreases with minibatch size, but guarantees that the optimization is still valid.

On the other hand, interleaving minibatch training with partial optimization of $\mathbf{w}$ (not all the way to optimality) makes the standard GAN training behave differently, however the exact properties of this process is hard to characterize and beyond the scope of this paper.

## B Proof of Claim 1

**Claim 1.** *The dual program to the minimization task*

$$\min_{\mathbf{w}} \quad \frac{C}{2} \|\mathbf{w}\|_2^2 + \frac{1}{2n} \sum_i \log(1 + \exp(-\mathbf{w}^\top \mathbf{x}_i)) + \frac{1}{2n} \sum_i \log(1 + \exp(\mathbf{w}^\top G_{\theta}(\mathbf{z}_i))).$$

*reads as follows:*

$$\max_{\lambda} \quad g(\theta, \lambda) = -\frac{1}{2C} \left\| \sum_i \lambda_{\mathbf{x}_i} \mathbf{x}_i - \sum_i \lambda_{\mathbf{z}_i} G_\theta(\mathbf{z}_i) \right\|^2 + \frac{1}{2n} \sum_i H(2n\lambda_{\mathbf{x}_i}) + \frac{1}{2n} \sum_i H(2n\lambda_{\mathbf{z}_i}),$$

$$\text{s.t.} \quad \forall i, \quad 0 \le \lambda_{\mathbf{x}_i} \le \frac{1}{2n}, \quad 0 \le \lambda_{\mathbf{z}_i} \le \frac{1}{2n}. \tag{5}$$

*with binary entropy $H(u) = -u \log u - (1-u) \log(1-u)$, and the optimal solution to the original problem $\mathbf{w}^*$ can be expressed with optimal $\lambda_{\mathbf{x}_i}^*$ and $\lambda_{\mathbf{z}_i}^*$ as*

$$\mathbf{w}^* = \frac{1}{C} \left( \sum_i \lambda_{\mathbf{x}_i}^* \mathbf{x}_i - \sum_i \lambda_{\mathbf{z}_i}^* G_\theta(\mathbf{z}_i) \right)$$

*Proof.* We introduce auxillary variables $\xi_{\mathbf{x}_i} = \mathbf{w}^\top \mathbf{x}_i$ and $\xi_{\mathbf{z}_i} = -\mathbf{w}^\top G_\theta(\mathbf{z}_i)$, the original minimization problem can then be transformed into the following equality constrained problem

$$\min_{\mathbf{w}} \quad \frac{C}{2} \|\mathbf{w}\|_2^2 + \frac{1}{2n} \sum_i \log(1 + e^{-\xi_{\mathbf{x}_i}}) + \frac{1}{2n} \sum_i \log(1 + e^{-\xi_{\mathbf{z}_i}})) \tag{6}$$

$$\text{s.t.} \quad \forall i, \quad \xi_{\mathbf{x}_i} = \mathbf{w}^\top \mathbf{x}_i, \quad \xi_{\mathbf{z}_i} = -\mathbf{w}^\top G_\theta(\mathbf{z}_i).$$

The corresponding Lagrangian has the following form

$$L(\mathbf{w}, \xi, \lambda, \theta) = \frac{C}{2} \|\mathbf{w}\|_2^2 + \frac{1}{2n} \sum_i \log(1 + e^{-\xi_{\mathbf{x}_i}}) + \frac{1}{2n} \sum_i \log(1 + e^{-\xi_{\mathbf{z}_i}}))$$

$$+ \sum_i \lambda_{\mathbf{x}_i} (\xi_{\mathbf{x}_i} - \mathbf{w}^\top \mathbf{x}_i) + \sum_i \lambda_{\mathbf{z}_i} (\xi_{\mathbf{z}_i} + \mathbf{w}^\top G_\theta(\mathbf{z}_i)) \tag{7}$$

Set the derivatives with respect to the primal variables to 0, we get

$$\frac{\partial L}{\partial \mathbf{w}} = C\mathbf{w} - \sum_i \lambda_{\mathbf{x}_i} \mathbf{x}_i + \sum_i \lambda_{\mathbf{z}_i} G_\theta(\mathbf{z}_i) = 0 \tag{8}$$

$$\frac{\partial L}{\partial \xi_{\mathbf{x}_i}} = -\frac{1}{2n} \frac{e^{-\xi_{\mathbf{x}_i}}}{1 + e^{-\xi_{\mathbf{x}_i}}} + \lambda_{\mathbf{x}_i} = 0 \tag{9}$$

$$\frac{\partial L}{\partial \xi_{\mathbf{z}_i}} = -\frac{1}{2n} \frac{e^{-\xi_{\mathbf{z}_i}}}{1 + e^{-\xi_{\mathbf{z}_i}}} + \lambda_{\mathbf{z}_i} = 0. \tag{10}$$

We can then represent the primal variables using the $\lambda$'s,

$$\mathbf{w} = \frac{1}{C} \left( \sum_i \lambda_{\mathbf{x}_i} \mathbf{x}_i - \sum_i \lambda_{\mathbf{z}_i} G_\theta(\mathbf{z}_i) \right) \tag{11}$$

$$\xi_{\mathbf{x}_i} = \log \frac{1 - 2n\lambda_{\mathbf{x}_i}}{2n\lambda_{\mathbf{x}_i}} \tag{12}$$

$$\xi_{\mathbf{z}_i} = \log \frac{1 - 2n\lambda_{\mathbf{z}_i}}{2n\lambda_{\mathbf{z}_i}}. \tag{13}$$

Eq.(9) and (10) also introduced extra constraints on $\lambda_{\mathbf{x}_i}$ and $\lambda_{\mathbf{z}_i}$, as follows

$$\forall i, \quad 0 \le \lambda_{\mathbf{x}_i} \le \frac{1}{2n}, \quad 0 \le \lambda_{\mathbf{z}_i} \le \frac{1}{2n}. \tag{14}$$

Substituting the primal variables back to the Lagrangian, we get the dual objective

$$g(\theta, \lambda) = \frac{C}{2} \left\| \frac{1}{C} \left( \sum_i \lambda_{\mathbf{x}_i} - \sum_i \lambda_{\mathbf{z}_i} \right) \right\|_2^2 - \frac{1}{2n} \log(1 - 2n\lambda_{\mathbf{x}_i}) - \frac{1}{2n} \log(1 - 2n\lambda_{\mathbf{z}_i})$$

$$+ \sum_i \lambda_{\mathbf{x}_i} \log \frac{1 - 2n\lambda_{\mathbf{x}_i}}{2n\lambda_{\mathbf{x}_i}} + \sum_i \lambda_{\mathbf{z}_i} \log \frac{1 - 2n\lambda_{\mathbf{z}_i}}{2n\lambda_{\mathbf{z}_i}} + \frac{1}{C} \left\| \sum_i \lambda_{\mathbf{x}_i} \mathbf{x}_i - \lambda_{\mathbf{z}_i} G_\theta(\mathbf{z}_i) \right\|_2^2$$

$$= -\frac{1}{2C} \left\| \sum_i \lambda_{\mathbf{x}_i} \mathbf{x}_i - \sum_i \lambda_{\mathbf{z}_i} G_\theta(\mathbf{z}_i) \right\|^2 + \frac{1}{2n} \sum_i H(2n\lambda_{\mathbf{x}_i}) + \frac{1}{2n} \sum_i H(2n\lambda_{\mathbf{z}_i}) \tag{15}$$

The overall dual problem is therefore

$$\max_\lambda \quad g(\theta, \lambda) = -\frac{1}{2C}\left\|\sum_i \lambda_{\mathbf{x}_i}\mathbf{x}_i - \sum_i \lambda_{\mathbf{z}_i}G_\theta(\mathbf{z}_i)\right\|^2 + \frac{1}{2n}\sum_i H(2n\lambda_{\mathbf{x}_i}) + \frac{1}{2n}\sum_i H(2n\lambda_{\mathbf{z}_i}),$$

$$\text{s.t.} \quad \forall i, \quad 0 \le \lambda_{\mathbf{x}_i} \le \frac{1}{2n}, \quad 0 \le \lambda_{\mathbf{z}_i} \le \frac{1}{2n}. \tag{16}$$

Once we have solved for the optimal $\lambda^*$, we can recover the optimal primal solution $\mathbf{w}^*$ using (11). $\qquad\square$

## C   Setting the step size $\Delta_k$ in the trust-region method

Pursuing this trust-region intuition, we can alternatively choose $\Delta_k$ based on the accuracy of the model $m_{k,\theta}(\mathbf{s})$. To this end it is often convenient to introduce the acceptance ratio

$$\rho = \frac{f(\mathbf{w}_k, \theta) - f(\mathbf{w}_k + \mathbf{s}, \theta)}{f(\mathbf{w}_k, \theta) - m_{k,\theta}(\mathbf{s})}, \tag{17}$$

which compares the real function value difference to the modeled one. If the acceptance ratio $\rho$ deviates significantly from 1 on either side, we may opt to decrease the trust region $\Delta_k$ and resolve the program given in Eq. (4) of the main paper, instead of accepting the step.

Intuitively, if $\rho$ specified in Eq. (17) is far from 1, the model function does not fit well the original objective. To obtain a better fit we resolve the program using a smaller trust region size $\Delta_k$.

## D   Proof of Claim 2

**Claim 2.** *The dual program to $\min_\mathbf{s} m_{k,\theta}(\mathbf{s})$ s.t. $\frac{1}{2}\|\mathbf{s}\|_2^2 \le \Delta_k$ with model function given as*

$$m_{k,\theta}(\mathbf{s}) = \frac{C}{2}\|\mathbf{w}_k + \mathbf{s}\|_2^2 + \frac{1}{2n}\sum_i \log\left(1 + \exp\left(-F(\mathbf{w}_k, \mathbf{x}_i) - \mathbf{s}^\top \frac{\partial F(\mathbf{w}_k, \mathbf{x}_i)}{\partial \mathbf{w}}\right)\right)$$

$$+ \frac{1}{2n}\sum_i \log\left(1 + \exp\left(F(\mathbf{w}_k, G_\theta(\mathbf{z}_i)) + \mathbf{s}^\top \frac{\partial F(\mathbf{w}_k, G_\theta(\mathbf{z}_i))}{\partial \mathbf{w}}\right)\right)$$

*is the following:*

$$\max_\lambda \quad \frac{C}{2}\|\mathbf{w}_k\|_2^2 - \frac{1}{2(C + \lambda_T)}\left\|-C\mathbf{w}_k + \sum_i \lambda_{\mathbf{x}_i}\frac{\partial F(\mathbf{w}_k, \mathbf{x}_i)}{\partial \mathbf{w}} - \sum_i \lambda_{\mathbf{z}_i}\frac{\partial F(\mathbf{w}_k, G_\theta(\mathbf{z}_i))}{\partial \mathbf{w}}\right\|_2^2$$

$$+ \frac{1}{2n}\sum_i H(2n\lambda_{\mathbf{x}_i}) + \frac{1}{2n}\sum_i H(2n\lambda_{\mathbf{z}_i}) - \sum_i \lambda_{\mathbf{x}_i}F_{\mathbf{x}_i} + \sum_i \lambda_{\mathbf{z}_i}F_{\mathbf{z}_i} - \lambda_T\Delta_k$$

$$\text{s.t.} \quad \lambda_T \ge 0 \quad \forall i, \quad 0 \le \lambda_{\mathbf{x}_i} \le \frac{1}{2n}, \quad 0 \le \lambda_{\mathbf{z}_i} \le \frac{1}{2n}.$$

*The optimal $\mathbf{s}^*$ to the original problem can be expressed through optimal $\lambda_T^*, \lambda_{\mathbf{x}_i}^*, \lambda_{\mathbf{z}_i}^*$ as*

$$\mathbf{s}^* = \frac{1}{C + \lambda_T^*}\left(\sum_i \lambda_{\mathbf{x}_i}^*\frac{\partial F(\mathbf{w}_k, \mathbf{x}_i)}{\partial \mathbf{w}} - \sum_i \lambda_{\mathbf{z}_i}^*\frac{\partial F(\mathbf{w}_k, \mathbf{z}_i)}{\partial \mathbf{w}}\right) - \frac{C}{C + \lambda_T^*}\mathbf{w}_k$$

*Proof.* In this optimization problem, the free variable is $\mathbf{s}$. We introduce short hand notations $F_{\mathbf{x}_i} = F(\mathbf{w}_k, \mathbf{x}_i), F_{\mathbf{z}_i} = F(\mathbf{w}_k, G_\theta(\mathbf{z}_i)), \nabla F_{\mathbf{x}_i} = \frac{\partial F(\mathbf{w}_k, \mathbf{x}_i)}{\partial \mathbf{w}}$ and $\nabla F_{\mathbf{z}_i} = \frac{\partial F(\mathbf{w}_k, G_\theta(\mathbf{z}_i))}{\partial \mathbf{w}}$. With these extra notations we can simplify the primal problem as

$$m_{k,\theta}(\mathbf{s}) = \frac{C}{2}\|\mathbf{w}_k + \mathbf{s}\|_2^2 + \frac{1}{2n}\sum_i \log\left(1 + e^{-F_{\mathbf{x}_i} - \mathbf{s}^\top \nabla F_{\mathbf{x}_i}}\right) + \frac{1}{2n}\sum_i \log\left(1 + e^{F_{\mathbf{z}_i} + \mathbf{s}^\top \nabla F_{\mathbf{z}_i}}\right)$$

$$\tag{18}$$

Again, we introduce auxillary variables $\xi_{\mathbf{x}_i} = \mathbf{s}^\top \nabla F_{\mathbf{x}_i}$ and $\xi_{\mathbf{z}_i} = -\mathbf{s}^\top \nabla F_{\mathbf{z}_i}$, and obtain the following constrained optimization problem

$$\min_{\mathbf{s},\xi} \quad \frac{C}{2}\|\mathbf{w}_k + \mathbf{s}\|_2^2 + \frac{1}{2n}\sum_i \log\left(1 + e^{-F_{\mathbf{x}_i} - \xi_{\mathbf{x}_i}}\right) + \frac{1}{2n}\sum_i \log\left(1 + e^{F_{\mathbf{z}_i} - \xi_{\mathbf{z}_i}}\right) \quad (19)$$

$$\text{s.t.} \quad \xi_{\mathbf{x}_i} = \mathbf{s}^\top \nabla F_{\mathbf{x}_i}, \quad \xi_{\mathbf{z}_i} = -\mathbf{s}^\top \nabla F_{\mathbf{z}_i}, \quad \forall i$$

$$\frac{1}{2}\|\mathbf{s}\|^2 \le \Delta_k$$

The corresponding Lagrangian is the following

$$L(\mathbf{w}, \xi, \lambda) = \frac{C}{2}\|\mathbf{w}_k + \mathbf{s}\|_2^2 + \frac{1}{2n}\sum_i \log\left(1 + e^{-F_{\mathbf{x}_i} - \xi_{\mathbf{x}_i}}\right) + \frac{1}{2n}\sum_i \log\left(1 + e^{F_{\mathbf{z}_i} - \xi_{\mathbf{z}_i}}\right)$$

$$+ \sum_i \lambda_{\mathbf{x}_i}(\xi_{\mathbf{x}_i} - \mathbf{s}^\top \nabla F_{\mathbf{x}_i}) + \sum_i \lambda_{\mathbf{z}_i}(\xi_{\mathbf{z}_i} + \mathbf{s}^\top \nabla F_{\mathbf{z}_i}) + \lambda_T\left(\frac{1}{2}\|\mathbf{s}\|^2 - \Delta_k\right) \quad (20)$$

Setting the derivatives of the primal variables with respect to the Lagrangian to 0, we get

$$\frac{\partial L}{\partial \mathbf{s}} = C(\mathbf{w}_k + \mathbf{s}) - \sum_i \lambda_{\mathbf{x}_i}\nabla F_{\mathbf{x}_i} + \sum_i \lambda_{\mathbf{z}_i}\nabla F_{\mathbf{z}_i} + \lambda_T \mathbf{s} = 0 \quad (21)$$

$$\frac{\partial L}{\partial \xi_{\mathbf{x}_i}} = -\frac{1}{2n}\frac{e^{-F_{\mathbf{x}_i} - \xi_{\mathbf{x}_i}}}{1 + e^{-F_{\mathbf{x}_i} - \xi_{\mathbf{x}_i}}} + \lambda_{\mathbf{x}_i} = 0 \quad (22)$$

$$\frac{\partial L}{\partial \xi_{\mathbf{z}_i}} = -\frac{1}{2n}\frac{e^{F_{\mathbf{z}_i} - \xi_{\mathbf{z}_i}}}{1 + e^{F_{\mathbf{z}_i} - \xi_{\mathbf{z}_i}}} + \lambda_{\mathbf{z}_i} = 0 \quad (23)$$

Therefore

$$\mathbf{s} = \frac{1}{C + \lambda_T}\left(\sum_i \lambda_{\mathbf{x}_i}\nabla F_{\mathbf{x}_i} - \sum_i \lambda_{\mathbf{z}_i}\nabla F_{\mathbf{z}_i}\right) - \frac{C}{C + \lambda_T}\mathbf{w}_k \quad (24)$$

$$\xi_{\mathbf{x}_i} = \log\frac{1 - 2n\lambda_{\mathbf{x}_i}}{2n\lambda_{\mathbf{x}_i}} - F_{\mathbf{x}_i} \quad (25)$$

$$\xi_{\mathbf{z}_i} = \log\frac{1 - 2n\lambda_{\mathbf{z}_i}}{2n\lambda_{\mathbf{z}_i}} + F_{\mathbf{z}_i}, \quad (26)$$

which includes the equation for $\mathbf{s}^*$.

Next we substitute these back to the Lagrangian to obtain the dual objective. We introduce another short hand notation $\square = \sum_i \lambda_{\mathbf{x}_i}\nabla F_{\mathbf{x}_i} - \sum_i \lambda_{\mathbf{z}_i}\nabla F_{\mathbf{z}_i}$, then $\mathbf{s} = \frac{1}{C+\lambda_T}\square - \frac{C}{C+\lambda_T}\mathbf{w}_k$, and the dual

objective can be written as

$$
\begin{aligned}
g(\lambda) \;=\;& \frac{C}{2}\left\|\frac{1}{C+\lambda_T}\left(\lambda_T \mathbf{w}_k + \square\right)\right\|^2 - \frac{1}{2n}\sum_i \log(1 - 2n\lambda_{\mathbf{x}_i}) - \frac{1}{2n}\sum_i \log(1 - 2n\lambda_{\mathbf{z}_i}) \\
&+ \sum_i \lambda_{\mathbf{x}_i}\left(\log\frac{1 - 2n\lambda_{\mathbf{x}_i}}{2n\lambda_{\mathbf{x}_i}} - F_{\mathbf{x}_i} - \frac{1}{C+\lambda_T}\left(\square - C\mathbf{w}_k\right)^\top \nabla F_{\mathbf{x}_i}\right) \\
&+ \sum_i \lambda_{\mathbf{z}_i}\left(\log\frac{1 - 2n\lambda_{\mathbf{z}_i}}{2n\lambda_{\mathbf{z}_i}} + F_{\mathbf{z}_i} + \frac{1}{C+\lambda_T}\left(\square - C\mathbf{w}_k\right)^\top \nabla F_{\mathbf{z}_i}\right) \\
&+ \frac{\lambda_T}{2}\left\|\frac{1}{C+\lambda_T}\left(\square - C\mathbf{w}_k\right)\right\|^2 - \lambda_T \Delta_k \\
=\;& \frac{1}{2n}\sum_i H(2n\lambda_{\mathbf{x}_i}) + \frac{1}{2n}\sum_i H(2n\lambda_{\mathbf{z}_i}) - \sum_i \lambda_{\mathbf{x}_i}F_{\mathbf{x}_i} + \sum_i \lambda_{\mathbf{z}_i}F_{\mathbf{z}_i} - \lambda_T\Delta_k \\
&+ \frac{C}{2(C+\lambda_T)^2}\|\lambda_T \mathbf{w}_k + \square\|^2 - \frac{1}{C+\lambda_T}(\square - C\mathbf{w}_k)^\top \square + \frac{\lambda_T}{2(C+\lambda_T)^2}\|\square - C\mathbf{w}_k\|^2 \\
=\;& \frac{1}{2n}\sum_i H(2n\lambda_{\mathbf{x}_i}) + \frac{1}{2n}\sum_i H(2n\lambda_{\mathbf{z}_i}) - \sum_i \lambda_{\mathbf{x}_i}F_{\mathbf{x}_i} + \sum_i \lambda_{\mathbf{z}_i}F_{\mathbf{z}_i} - \lambda_T\Delta_k \\
&- \frac{1}{2(C+\lambda_T)}\|\square - C\mathbf{w}_k\|^2 + \frac{C}{2}\|\mathbf{w}_k\|^2 \qquad\qquad (27)
\end{aligned}
$$

which is exactly the dual objective in the claim.  $\square$

## E More Experiment Details

### E.1 Toy dataset

The toy 2D dataset used in the paper consists of a mixture of 5 2D Gaussian components, the Gaussians have covariance matrix of $0.1I$ with means being uniformly spaced on a circle of radius 2.

Here we present additional results on an extra 8-mode dataset, where each of the 8 components in the 8-mode dataset is a Gaussian distribution with a covariance matrix of $0.02I$, and again the means of the components are arranged on a circle of radius 2. We use both datasets to investigate properties such as low probability regions and low separation of modes.

To train the linear GAN we employ RBF features based on a set of anchor points $\{x_1, ..., x_n\}$, then for an arbitrary $x$, the features for $x$ is computed as the following

$$
\phi(x) = \left[\frac{\exp(-\frac{1}{T}\|x - x_1\|^2)}{Z}, ..., \frac{\exp(-\frac{1}{T}\|x - x_n\|^2)}{Z}\right]^\top, \quad \text{where} \quad Z = \sum_i \exp(-\frac{1}{T}\|x - x_i\|^2).
$$

We set $T$ to 0.2 for all experiments.

Experiment results are shown in Fig. 1.

### E.2 MNIST Model Details

We used a generator architecture similar to that in [17].Instead of directly projecting the initial hidden variables to $4 \times 4$ images, we first feed it through a fully-connected hidden layer. In the intermediate layers, we use $4 \times 4$ upconvolution kernels with stride 2, ReLU activation, and batch normalization. At the output layer, we fed it through 1 extra $3 \times 3$ convolution layer without changing the image size. Instead of Tanh output activation, we use a Sigmoid function, and our data takes pixel value between 0 and 1. For all of our experiments, our initial hidden dimension is 32. For discriminator, our pretrained MNIST convnet uses 3 3x3 convolution layers with max pooling and ReLU activation, and 2 fully connected hidden layers with ReLU activation as well.

|  | Ours | Standard GAN |
|---|---|---|

Figure 1: Original data (blue) and samples obtained from the learned generator (green) for our approach (left) and standard GAN (right). We show results for 8-mode (top) and 5-mode data (bottom). For each of the approaches we demonstrate usage of RBF features.

Figure 2: Nonlinear discriminator experiments on CIFAR-10, learning curves and samples organized by class.

## E.3 CIFAR-10

For the generator, we use an architecture similar to the one described for MNIST. For the discriminator, it is similar to MNIST as well, except before each max pooling operation there are 2 convolutional and ReLU layers instead of 1. The width of the network here is also greater than the one for MNIST experiment.

We provide in Fig. 2 the primal objective, the discriminator accuracy and the model function value (primal approx.) throughout training, and top samples for each class. From the training curves we see that the proposed trust-region cost-linearization technique is significantly more stable than either the score linearization or standard GANs. The score linearization method does a better job approximating the discriminator at the begining, but then suffered from bad solution to the dual problem given by the Ipopt solver.

## Footnotes

*Now at DeepMind, yujiali@google.com