[Reviews · NeurIPS 2017]

Reviewer 1



To avoid the instability of the training curves present when training GANs generally, the paper proposes to solve the dual optimization problem for the discriminator rather than the primal problem, which is the standard GAN's formulation. The results show that dualizing the discriminator yields to monotonous training curves, which is not the case when considering the standard formulation where the training curves are spiky. Adopting the standard GAN approach, which is well-known in the literature, the paper proposes a new interesting way to solve the GAN's learning problem. The idea of dualizing the optimization problem is well motivated since it's fairly natural. Moreover, the proposed method is mathematically relevant and clearly validated by experiments. The paper is well written and the adopted formalism is consistent. Furthermore, the supplementary materials detail the experiments and the proofs of the main results, which are important to the paper's understanding. I have just a little remark concerning the figure 3 of the main paper: after convergence, it seems that Dualing-GAN may introduce some variance in the estimation of the real data distribution. In fact, the variance per mode of the generated samples, using Dualing-GAN, seems to be bigger than the one, using standard GAN. I recommend the authors to discuss this point to make sure that there is no negative impact on the results in general.

Reviewer 2



This submission proposes an alternative view on GAN training by replacing the inner learning of the discriminator by its dual program. In the case of linear discriminators this leads to a stable and simpler maximization problem. The authors propose strategies to cope with non-linear discriminators, by either taking linear approximations to the cost function, or to the scoring function (typically the CNN in the case of image GANs). PROS: - The paper proposes an alternative and simple view on GAN training. - The paper reads well and the formalism is well chosen. - The proposed algorithm (with scoring function linearization) leads to interesting initial results on CIFAR and MNIST. CONS: - The proposed approach requires defining a dual variable for each sample z_i. Therefore, the dual program needs to be solved to satisfactory precision at each step. - If I am not mistaken, Claim 1 is simply the dual of a binary logistic linear classification problem. Then, it might be good to present it that way, and avoid calling that a "claim" as this is a classical result. - From a purely formatting point of view, the graphical presentation of results could be improved, as it is hard to read. Moreover, only several runs, and the corresponding learning curves are given. Would it be possible to give other quantitative measures? How often does GAN training diverge? Is the proposed algorithm helping that metric? Overall, I think that this paper gives a simple view on GAN training using the dual of the inner problem. I enjoyed reading the paper and both the formal and experimental results seem correct. Because of this I lean towards acceptance - however, since I am not a pure GAN expert, I may have misevaluated the importance of this work.

Reviewer 3



This paper proposes a new method to train GAN: firstly it assumes the discriminator is linear, then the GAN model is solved by dual optimization, secondly since the discriminator is non-linear, to train the GAN model using dual optimization, local linearization for the cost or score function is repeated used. Experiments show the effectiveness of the proposed approach. The innovation of in the scenario of optimization, is moderate. It is natural to derive corresponding dual optimization for the case of linear discriminator which has limited representation power for practical application as well as the case of local linearization. The experiments are not strong enough, specially the so-called instability problem is still not overcome or mitigated. The author should better illustrate the advantage of the proposed approach through the experiments.